# Orientational Preferences of GPI-Anchored Ly6/uPAR Proteins

**DOI:** 10.3390/ijms24010011

**Published:** 2022-12-20

**Authors:** Maxim M. Zaigraev, Ekaterina N. Lyukmanova, Alexander S. Paramonov, Zakhar O. Shenkarev, Anton O. Chugunov

**Affiliations:** 1Shemyakin-Ovchinnikov Institute of Bioorganic Chemistry, Russian Academy of Sciences, Miklukho-Maklaya str. 16/10, 119997 Moscow, Russia; 2Phystech School of Biological and Medical Physics, Moscow Institute of Physics and Technology, Institutskiy per. 9, 141701 Moscow, Russia; 3Faculty of Biology, MSU-BIT Shenzhen University, Shenzhen 518172, China; 4Interdisciplinary Scientific and Educational School of Moscow University “Molecular Technologies of the Living Systems and Synthetic Biology”, Biological Faculty, Lomonosov Moscow State University, Leninskie Gory, 119234 Moscow, Russia; 5International Laboratory for Supercomputer Atomistic Modelling and Multi-Scale Analysis, National Research University Higher School of Economics, Myasnitskaya str. 20, 101000 Moscow, Russia

**Keywords:** Ly6 proteins, Ly6/uPAR, three-finger proteins, GPI-anchored proteins, N-glycans, O-glycans, molecular dynamics, orientational analysis, protein–membrane interactions

## Abstract

Ly6/uPAR proteins regulate many essential functions in the nervous and immune systems and epithelium. Most of these proteins contain single β-structural LU domains with three protruding loops and are glycosylphosphatidylinositol (GPI)-anchored to a membrane. The GPI-anchor role is currently poorly studied. Here, we investigated the positional and orientational preferences of six GPI-anchored proteins in the receptor-unbound state by molecular dynamics simulations. Regardless of the linker length between the LU domain and GPI-anchor, the proteins interacted with the membrane by polypeptide parts and N-/O-glycans. Lynx1, Lynx2, Lypd6B, and Ly6H contacted the membrane by the loop regions responsible for interactions with nicotinic acetylcholine receptors, while Lypd6 and CD59 demonstrated unique orientations with accessible receptor-binding sites. Thus, GPI-anchoring does not guarantee an optimal ‘pre-orientation’ of the LU domain for the receptor interaction.

## 1. Introduction

Ly6/uPAR proteins are valuable players in the physiology of animals, including humans [1,2,3]. They have been found in *Echinodermata* [4,5], *Platyhelminthes* [6], *Arthropoda* [7], and *Chordata* [1]. The name of the Ly6/uPAR superfamily comes from two representatives: lymphocyte antigen-6 (Ly6) [8] and urokinase-type plasminogen activator receptor (uPAR, also called PLAUR) [9]. The human genome contains 35 Ly6/uPAR genes, while the mouse genome—61 [10]. This superfamily is characterized by the LU domain (60–90 amino acids) composed of a compact disulfide-stabilized β-structural core (known as a ‘head’) and three protruding loops (loops I–III, known as ‘fingers’) [1,11]. Ly6/uPAR members are often referred to as three-finger proteins or Ly6 proteins. In mammals, Ly6/uPAR representatives typically contain only one LU domain, with the exception of Lypd3, Lypd5, and CD177, containing two, and uPAR, containing three LU domains [2,10]. Snakes are also known to contain a range of Ly6/uPAR proteins, including three-finger toxins, acetylcholinesterase inhibitors, and anticoagulants containing one LU domain [12,13,14], and phospholipase A2 inhibitors, probably containing two LU domains [15]. The LU domains can also be parts of larger proteins; for example, they are found in the extracellular fragments of the transforming growth factor beta receptor (TGFBR) family, including TGFBR1, TGFBR2, bone morphogenetic protein receptors, and activin receptors [2].

On the basis of subcellular localization, Ly6/uPAR proteins are classified as glycosylphosphatidylinositol (GPI)-anchored to the cell membrane (e.g., Lynx1, Lynx2 (also known as Lypd1), Lypd6, Lypd6B, Ly6H, and Ly6G6E) [2,16] or secreted (e.g., SLURP-1 [17], SLURP-2 [18], and snake three-finger toxins [12,13,14]). Some proteins (e.g., CD59, PSCA, and uPAR) can co-exist in both forms [1]. In addition, many Ly6 proteins are heavily glycosylated. Ly6/uPAR proteins are involved in the regulation of neuronal activity, cell proliferation, migration, cell–cell interaction, immune cell maturation, macrophage activation, cytokine production, the progression of inflammation, complement activity, angiogenesis, wound healing, and embryogenesis [1,2]. In mammals, they are found in the nervous, endocrine, reproductive, and immune systems, blood cells and in the epithelium [10]. Some of the GPI-anchored Ly6 proteins regulate the signaling of nicotinic acetylcholine receptors (nAChRs) (Lynx1, Ly6H, Lynx2, Lypd6, and Lypd6B) [1,3]. However, others may have alternative molecular targets, e.g., CD59 interacts with C8/C9 complement proteins [19,20], and Lynx1 modulates muscarinic acetylcholine receptors (mAChRs) [21]. In addition to nAChRs, Lypd6 targets the Wnt receptor Frizzled8 and the Wnt coreceptor low-density lipoprotein receptor-related protein 6 (LRP6) [22,23].

Despite the possible important role of GPI-anchor and glycosylation in the interaction of Ly6/uPAR proteins with their targets, these post-translational modifications have been previously considered in very few works [24]. GPI-anchor imposes significant configurational restraints on the behavior of Ly6 proteins and, thus, can be vital for their proper functioning. Here, without any pre-assigned hypothesis, we studied an intrinsic pre-orientation relative to the cell membrane and possible positional restraints imposed by the GPI-anchor on a set of six GPI-anchored proteins having a single LU domain in a receptor-unbound state. Five of them (Lynx1 [21,25,26,27], Ly6H [28,29,30], Lynx2 [31,32,33], Lypd6 [34,35,36,37], and Lypd6B [35,38,39,40]) regulate nicotinic cholinergic signaling in the nervous system, while CD59 protects the body’s own cells from complement-mediated lysis [19,20,41]. We built the most exhaustive to date models of Ly6 proteins based on the available 3D structures (Appendix A); added post-translational modifications, including the GPI-anchors and glycans (Appendix A); immersed them into lipid membranes of a different composition; and calculated 2–3 μs of all-atom molecular dynamics (MD) for each system (Appendix A). Then, we determined the following parameters: (1) the position of the center of mass (COM) of the protein relative to the membrane surface; (2) the orientation of the protein (in terms of tilt/rotation angles) relative to the membrane; and (3) the intermolecular contacts between the Ly6 protein and lipids. The data obtained will be useful in studying the interaction of receptors with their GPI-anchored ligands.

## 2. Results and Discussion

Representative positions and orientations of the GPI-anchored Ly6 proteins in the membrane are provided in Figure 1. Generally, both GPI lipid tails were anchored in the membrane, behaving similarly to the membrane lipid tails. At the same time, the polypeptide moieties of the Ly6 proteins floated at the bilayer surface, sometimes raising above it (Figure 2A). The *C*-terminal linkers of the proteins connecting the LU domains with the GPI-anchors, as well as the carbohydrate moieties of the GPI-anchors, underwent significant folding at the initial stage of MD. This process, probably related to an entropic coiling [42] of the linker chain, resulted in the protein association with the membrane surface, regardless of the linker length (provided in Table 1) (see Appendix A). The obtained results agree with the previously reported modeling of the GPI-anchored GFP, where protein contacts with the membrane were observed [43].

To extract functionally relevant motions from obtained Ly6 modulators’ trajectories, we performed principal component analysis (PCA). Protein backbone atoms were used for covariance matrix determination and eigen values calculation. PCA revealed that the extremal protein configurations for the most significant principal components represent the cases when proteins: (1) are located near or far from the membrane; (2) are inclined towards the membrane either by the first or third loop; and (3) are tilted to the membrane by a central loop or a ‘head’ (Appendix A). Thus, we introduced three parameters for a system state description: Z (the distance from the protein center of mass to the membrane), α (the tilt angle of the central loop to the membrane), and β (the rotation of the protein plane via the first or third loop to the membrane). These three principal components of proteins’ mobility seem to be enough for the coarse description of the anchored modulators’ movements because the internal folding of their protein parts is negligible as compared to the “rise”, “pitch”, and “roll” movements depicted by the Z, α, and β parameters, respectively.

### 2.1. Centers of Mass (COM) Positions

COM positions relative to the membrane surface are presented as distributions (Figure 2A) and the respective median values (Table 1). Three numbers are usually used to describe the position of the COM: x-, y-, and z-components. However, due to the translational symmetry with respect to the membrane (namely, x- and y-components), only the z-component is considered for the analysis.

We found that the distribution of the COM position may be either unimodal (CD59, Lynx1, Lynx2, and Lypd6B) or bimodal (Ly6H and Lypd6). Moreover, the median COM positions differ significantly for different proteins, indicating the adoption of different modes of interaction with the membrane. The largest difference can be observed for CD59 and Lypd6B (Figure 2A and Figure 3A). CD59 ‘lies’ on the membrane surface, while Lypd6B was lifted and based on N-glycan and loop III, which prevented tight association of the protein with the membrane. This mode of the Lypd6B/membrane interaction was stabilized by the sporadic interactions of N-glycan with the GPI-anchor, the interaction of the long *N*-terminal protein region (20 residues, which precede the LU domain, Table 1) with the membrane, the *C*-terminus of the protein, and the extramembrane part of the GPI-anchor (see Appendix A).

The lifted position of Lypd6B relative to CD59 could result from the significantly longer *C*-terminal linker (24 vs. 8 residues, Table 1), but we did not observe a good correlation between the lengths of the *C*-terminal linkers and the median COM positions, which do not exceed 2.3 nm (Pearson’s correlation coefficient R = 0.45, see Appendix A). This emphasizes a localization of the Ly6 proteins near the membrane, rather than remotely, as might be expected for the long-linker proteins (Lynx2, Lypd6, and Lypd6B). Nevertheless, Lynx2 significantly rose above the membrane (COM position ~ 4.5–5.0 nm) approximately in the middle of the MD trajectory (1300–1400 ns of the total 2000 ns) and subsequently descended to the membrane (see Appendix A). Probably, similar rises occur also for Lypd6 and Lypd6B, but they were not observed in our trajectories, probably due to insufficiently long MD and, consequently, an incomplete scanning of the conformational space.

Bimodal COM distributions, observed only for Ly6H and Lypd6, indicate that the *C*-terminal linker length (5 and 20 residues, respectively) does not correlate with the conformational freedom of the protein. For Lypd6, we observed two unequally populated peaks: the major one at 1.7 nm and weak—at ~2.2 nm (Figure 2A). This may indicate the ability of GPI-anchored Lypd6 to interact with several membrane targets of different “heights”: nAChRs [35] and the Wnt coreceptor LRP6 [23]. Two equally populated peaks were observed in the COM position distribution for Ly6H. The peaks at 1.8 and 2.3 nm form a relatively wide plateau (Figure 2A), indicating very high conformational flexibility of the Ly6H protein despite the very short *C*-terminal linker. We can speculate that Ly6H also may possess an additional (not yet identified) molecular target besides nAChRs.

### 2.2. Orientational Preferences

To define object orientation completely, usually three rotational angles are required. However, there is a rotational symmetry for GPI-anchored proteins around normal to membrane $\vec{Z}$—for this reason, only two rotational angles are enough to identify orientation: tilt (α) and rotation (β). They were calculated through the two mutually perpendicular vectors $\vec{A}$ and $\vec{B}$, lying in the β-sheet plane (see insets in Figure 2 and Appendix A). $\vec{A}$ is directed from the ‘head’ of the protein to the tip of the central loop (II) and runs along the conserved antiparallel β-sheet. The tilt angle (α = 90° − ∠($\vec{A}$, $\vec{Z}$), α ∈ [−90°, +90°], where $\vec{Z}$ is a membrane normal) determines, which part of the protein tilts towards the membrane: the ‘head’ (α > 0°) or the central loop (α < 0°); a zero value corresponds to the central loop being parallel to the membrane. Vector $\vec{B}$ is directed from loop III to loop I and describes the transverse tilt (relative to $\vec{A}$) of the molecule. The rotation angle (β ∈ [−180°, +180°]; β = 90° − ∠($\vec{B}$, $\vec{Z}$) if F_z_ ≥ 0; β = 90° + ∠($\vec{B}$, $\vec{Z}$) if B_z_ ≥ 0 and F_z_ < 0; or β = −270° + ∠($\vec{B}$, $\vec{Z}$) if B_z_ < 0 and F_z_ < 0; B_z_ and F_z_ are projections of $\vec{B}$ and $\vec{F}$ on $\vec{Z}$, respectively) indicates that the protein is oriented towards the membrane either by loop I (β ~ −90°) or loop III (β ~ +90°); β = 0° or ±180° means that a protein’s β-sheet is parallel to the membrane, and loops I and III are equidistant from the membrane. β = 0°s means that both *N*- and *C*-termini face the membrane (the ‘ventral’ side of the β-structure is down); β = ±180° corresponds to the case where the ‘dorsal’ side of the β-structure is down. The one- and two-dimensional histograms of the probability distribution of α/β angles are shown in Figure 2B,C and Figure 4, respectively. The extremal orientations of the GPI-anchored Ly6 proteins in the membrane are provided in Figure 3B,C.

Negative α values predominate for Lynx1, Lynx2, Lypd6B, and Ly6H (Figure 2B, Table 1), which suggests that most of the studied Ly6 proteins in the receptor-unbound state prefer touching the membrane by the ‘fingers’. The largest negative tilt (α = −82.3°) was observed for Ly6H (Figure 3B, *left*), where it was stabilized by the intensive polar and hydrophobic interactions of the loop II tip and N-glycan with the membrane lipids (Table 2). Interestingly, bimodal distributions of the tilt angle α with positive and negative values were observed for Ly6H and Lynx2 (Figure 2B). This indicates that even proteins with short *C*-terminal linkers (5 and 11 residues, respectively) can switch their ‘fingers’ orientation from down to up.

In contrast, CD59 had a weak positive tilt (α = 17.8 ± 9.0°), and this configuration was quite stable during MD due to the specific conformation of the *C*-terminal linker (Figure 3A *left*, shown in red), which went approximately parallel to the membrane surface and prevented a detachment of the protein’s ‘head’ from the membrane and tilting of the loop tips toward the membrane surface (Appendix A).

The pronounced positive shift of the Lypd6 tilt angle α (Figure 2B) can be explained by the extensive interactions of the protein’s ‘head’ and the highly mobile *N*-terminal region with the membrane, which lift the Lypd6 ‘fingers’ up (Figure 3B, *right*; Table 2; Appendix A). The significant difference of the Lypd6 and Lypd6B orientations is quite unexpected because their LU domains are ≈60%similar. The difference probably comes from the distinct distribution of the charged residues in the protein ‘heads’ [11], contributing to dissimilar pharmacology [23,35].

Distributions of the rotation angle β revealed that all studied Ly6 proteins (except CD59) tend to orient loop I rather than loop III towards the membrane (β < 0°; Figure 2C; Table 1). However, sporadic “positive transitions” were observed for Lynx2 and Lypd6. The observed extreme modes of the rotation (β ≈ ±90°) are illustrated by the Lynx1 (β_min_ = −82.1°) and Lynx2 (β_max_ = 89.6°) proteins (Figure 3C). Among the proteins studied, only CD59 and Lynx2 tended to orient their ‘ventral’ side towards the membrane (β ≈ 0°), while Lypd6 faced the membrane by the ‘dorsal’ side (β ≈ 180°). Interestingly, other proteins (Ly6H, Lynx1, and Lypd6) switched between the ‘ventral’ and ‘dorsal’ orientations (Figure 2C).

The CD59 orientation differed significantly from the other Ly6 proteins. It demonstrated narrow and distinctive distributions of the COM position and tilt/rotation angles (Figure 2). This may highlight a different target of CD59 [19,20,41] as compared to the nAChR ligands (Lynx1, Lynx2, Lypd6, Lypd6B, and Ly6H), despite the generally conserved three-finger fold.

The probability distribution histograms for different tilt and rotation angles for all the proteins studied are shown in Figure 4; these data represent the free energy landscape in two principal coordinates (α; β) and the general preference of different orientations. As one can see, although generally very similar and anchored in the same way, the studied Ly6 proteins exhibit individual behavior.

### 2.3. Interaction of Ly6 Proteins with Membrane Lipids

Analysis of the MD trajectories revealed long-lived contacts with the membrane lipids not only for the residues of GPI-anchor, but also for protein amino acid residues and glycans (Table 2 and Appendix A). Various types of contacts were found: ion–ion and ion–dipole interactions, hydrogen bonds, π–cation, and hydrophobic interactions (see Appendix A). There was a large variation in the number of long-lived ionic contacts, from two for Lypd6B to seven (two of them from the long N-terminal region) for Lypd6. Interestingly, Lynx1—the only non-glycosylated protein in this study—formed a significantly greater number of stable polar contacts (ionic and hydrogen bonds) with the membrane than other proteins, 15 vs. 8–11, respectively (see Table 2). Probably, the absence of glycans ensures a tighter interaction of the polypeptide part of Lynx1 with the lipids. Besides ionic and polar contacts, amino acid residues can simultaneously form up to 6–7 hydrophobic contacts with lipids (see Appendix A).

Glycans also interacted with the membrane but not so persistently as amino acids due to their high mobility. For Lynx2, Ly6H, Lypd6B, and CD59, the stable hydrogen bonds and hydrophobic contacts between N- and O-glycans and the membrane lipids were observed. Glycans do not interact with the membrane only in the case of Lypd6. In addition, different GPI-anchors’ residues contacted the membrane in all studied systems. Contacts were observed for both phosphatidylinositol (DSPI-1, SAPI-1) residues directly incorporated into the membrane and for carbohydrate residues, including Man-5 and Man-6, furthest from the membrane (Table 2). Membrane-embedded parts of the GPI-anchor can form up to 25–30 hydrophobic contacts due to the interaction with acyl lipid chains.

The largest number of hydrophobic contacts with lipids was observed for CD59. The contacts were formed by the residues of the protein ‘head’, loop III, C-terminal linker, and O-glycan (Table 2). Interestingly, while polar contacts predominated in the interaction of other Ly6 proteins with the membrane, hydrophobic contacts predominated for CD59.

According to the ‘membrane catalysis’ concept [44,45], the binding of a ligand to the membrane can optimize ligand–receptor interactions. The attachment of the GPI-anchor to a soluble three-finger domain could have the following consequences: (1) the partition of a ligand into the appropriate membrane compartment in the vicinity of the target receptor, and (2) the ‘pre-orientation’ of the three-finger domain that carries the receptor-binding site for optimal interaction with the receptor.

Among the studied Ly6 proteins, the positions of the receptor-binding sites were established only for Lynx1/nAChR [46], Lypd6/LPR6 [23], and the CD59/membrane attack complex [20]. Moreover, most of the Ly6 proteins studied to date (except CD59) interact with target receptors by the loop regions [12,13,17,23,46]. In our MD trajectory, the loop II of Lypd6, containing Asn88-Ser-Ile90 motif responsible for the interaction with LPR6 [23], was lifted high enough above the membrane surface to interact with the receptor (Figure 1D). Thus, the LU domain of Lypd6 is probably pre-oriented for effective receptor binding. A similar situation was observed for CD59: its receptor-binding site lies on the ’dorsal’ side of the LU domain near Trp65 and is accessible in the membrane-bound protein. At the same time, some of the CD59 residues located on the edge of the receptor-binding site (e.g., Phe67 and Asn73) contact the lipids (Table 2). Nevertheless, we assume that CD59 is also ‘pre-oriented’ for receptor binding.

In contrast, the Lynx1 loop II residues participating in the nAChR binding [46] form strong contacts with the membrane lipids (Table 2, Figure 1A). The other Ly6 proteins acting on nAChRs (Lynx2, Lypd6B, and Ly6H) also tended to interact with the membrane by the tip of loop II and loop I (α < 0°, β < 0°). Thus, the LU domains of these proteins are not ‘pre-oriented’ for optimal receptor binding and should raise their ‘fingers’ to interact with the ligand-binding site at the receptor.

### 2.4. Data Relevance and Application to In Vivo

Our in silico study provides data on spatial position for a range of three-finger proteins regarding membrane surfaces. Despite the existence of several powerful experimental methods, it is difficult to obtain such data in vivo or in model systems. For example, to determine the position and orientation of the protein above the membrane by the fluorescence or EPR spectroscopies, the introduction of several fluorescent or paramagnetic labels into the protein and some labels in the membrane is needed. In this case, the labels (that are usually large) can significantly disturb the position of the protein and membrane properties. On the other hand, the systems with Ly6/uPAR proteins in the model membranes (e.g., in liposomes) are too large for solution NMR studies, and they are insufficiently ‘solid’ and ordered to be studied by solid-state NMR. The other method, which became popular in the last few years—cryo-electron microscopy—is also not applicable. Ly6/uPAR proteins are too small for EM studies. Thus, in silico simulations are practically the only method to obtain information about the behavior of such dynamic systems as GPI-anchored proteins.

We performed our calculations using an explicit solvent and membrane model. However, it would be reasonable to note that GPI-anchored proteins in vivo are surrounded by a complex mixture of various components, including water-soluble and membrane proteins, which are able to influence their position and orientation, as well as their contact with lipids. Here, our system can be considered as a model like those used in experimental methods to simplify the study. For example, the interaction of GPI-anchored Ly6/uPAR proteins (e.g., Lynx1 and Lypd6) with nicotinic acetylcholine receptors (nAChRs) cannot be studied in vivo due to the very heterogeneous environment in the different tissues of the organism. On the other hand, in vitro studies such as electrophysiology measurements are usually done in a controlled environment represented by a buffer applied on the individual cell or cell-patch through the perfusion system. This means that the receptors and GPI-anchored proteins under study do not contact the different molecules presented in the ‘biological’ fluids but are submerged in the controlled solution represented by water and salts. However, the validity of electrophysiology in vitro studies is usually not questioned. At the same time, an attempt to introduce additional interactions into the MD simulations does not seem to provide additional significant information to the information obtained in our case due to the wide variety of external conditions and the locations of the GPI-anchored Ly6/uPAR proteins. For example, the in silico simulation of Ly6/uPAR proteins in a concentrated solution of acetylcholine or glutamate (the conditions sometimes occurring in the synapse) does not give adequate information about the behavior of the proteins in the lung or skin or even in the synaptic cleft in the absence of a neurotransmitter.

Additionally, although relatively long, our simulation lengths definitely cannot be considered exhaustive, so it should be noted that MD pictures are almost always just a glance through a keyhole at how molecules actually behave.

## 3. Materials and Methods

### 3.1. Systems Preparation

To perform all-atom MD simulations, we set up series of systems containing Ly6 modulators, considering GPI-anchors, glycosylation, and an explicit membrane/water environment. Systems were built using the CHARMM-GUI software package [47] using the instruments: *Membrane Builder*, *Glycolipid Modeler* [48], and *Glycan Modeler* [49]. Models of full-size, human, mature Ly6 proteins, containing all *N*- and *C*-terminal amino acids, which are usually absent from experimental structures (although necessary for GPI-anchor attachment), were taken from the AlphaFold database [50,51] (Appendix A). Their RMSD values from the respective experimental structures were all below 1.5 Å, which is in the range of normal RMSD values observed in the MD trajectories, except for Ly6H, which still had no structure determined.

To estimate the pK_a_ values of amino acids within the studied Ly6 proteins, we used the PROPKA prediction program [52,53]. We obtained pK_a_ values corresponding to a mainly deprotonated state for Asp, Glu, and N-termini; a mainly protonated state for Arg, Lys, Cys, and Tyr; and a mainly deprotonated state for most of the His residues. The maximal histidine pK_a_ predicted values were 7.05 (for His-69 in CD59) and 7.00 (for His-37 in Lypd6). Concerning this, we uniformly set all histidines as deprotonated. In our case, all C-termini were deprotonated due to amide bond formation with D-glucosamine residue of the GPI-anchor. Detailed output values from PROPKA predictions can be found in Appendix A.

Many Ly6 proteins are glycosylated, which is frequently omitted in modeling studies. To model the *N*-, *O*-glycans, and GPI-anchor of CD59, we searched the experimental MALDI-MS and HPLC data and chose the most frequent isoforms [54,55,56,57]. Because the exact *N*-glycans structure of other Ly6 proteins is currently unknown, we used an isoform widespread in the central nervous system (CNS), according to the *N*-glycome data in mice [58]. To model the GPI-anchors of Lynx1, Lynx2, Lypd6, and Lypd6B, the structure of the most common isoform in the CNS was used, although their exact structures are also unknown [59].

To build the CD59 system, we used a model bilayer (Appendix A), resembling an erythrocyte outer layer, where normally this protein resides [60]. Other systems had a lipid composition characteristic for rafts in neuronal membranes [24,61] (Appendix A).

Detailed structures of GPI-anchors and glycans are described in Appendix A. Standard CHARMM36m parametrization supplied by the CHARMM-GUI server was used to describe GPI-anchor [48] and glycan [62,63] behavior.

### 3.2. Molecular Dynamics Simulations

MD trajectories were calculated in all-atom CHARMM36m forcefield [64] and TIP3P water model via GROMACS [65]. For a more accurate description of π–cation interactions, an additional set of CHARMM36-WYF parameters [66] was used. MD calculations included the following stages: energy minimization (steepest descent algorithm), NVT relaxation (250 ps), NPT relaxation with C-rescale barostat [67] (2000 ps), and production MD (2–3 μs).

The production MD calculations for equilibrated systems were performed in an NPT ensemble at 310 K with a V-rescale thermostat [68] and a Parinello–Raman barostat [69] with a time step of 2 fs. No position restraints were applied to any molecules during the production MD phase, so the structure of proteins and their GPI-anchors and glycans was allowed to relax and change.

The details on built systems (box sizes, number of lipid and water molecules) and the total lengths of the calculated trajectories are given in Appendix A.

### 3.3. Data Analysis

To perform principal component analysis (PCA), we utilized GROMACS utilities *gmx trjconv*, *covar*, and *anaeig*. Prior to analysis, the modulator trajectories with protein backbone fitted to x- and y- but not z-components of the box were obtained (-*fit rotxy+transxy* option in *gmx trjconv*). Then, covariance matrices were constructed using *gmx covar* performed with -*nofit* option. For eigenvalue and eigenvector determination, we took advantage of the *gmx anaeig* procedure; protein backbone atoms were used as input for eigenvector calculation.

To analyze the center of mass (COM) position and the orientation of the proteins during MD, we used GROMACS utilities *gmx trjconv*, *make_ndx*, *trjcat*, and in-house Python scripts, which use NumPy and Matplotlib libraries.

To determine the protein COM position relative to the membrane, we used the *gmx make_ndx* procedure to define two index groups: amino acids and phosphorus atoms of upper lipid monolayer. Using the *gmx trjcat* procedure, we extracted the COM coordinates for both groups. The difference in the Z coordinate provides the position of protein COM relative to the membrane.

To determine protein orientation with respect to the membrane, we performed structure superposition using PyMOL and sequence alignment using Jalview [70] and CysBar [71]. On this basis, we then selected four groups of amino acid residues to establish orientation angles. The exact definitions and ways of calculation of orientational angles with respect to the membrane are described in Appendix A.

### 3.4. Lipid Contacts

For analysis of intermolecular contacts between the Ly6 proteins and membrane lipids, we utilized our in-house IMPULSE software package (*cont_stat.js* procedure) [72]. Hydrophobic contacts were determined according to the concept of molecular hydrophobic potential (MHP) [73].

A complete list of all found contacts is available in the Appendix A. The values in the table are the relative lifetimes of interactions for all types of lipids in total and separately (cholesterol, sphingomyelins, phosphatidylcholines, and phosphatidylethanolamines). Relative lifetime means the fraction of MD trajectory where the corresponding contact with lipid was observed: the value 0 corresponds to the complete absence of contact; the value 1 corresponds to the presence of one contact throughout the entire trajectory. If more than one contact of the corresponding protein group with the lipid was observed during MD, relative lifetime can exceed 1.

The following software versions were used: GROMACS 2021–2022 (different versions); CHARMM-GUI 3.7; IMPULSE 21.09; PROPKA 3.4; and PyMOL 2.5.4.

## 4. Conclusions

In summary, the behavior of GPI-anchored Ly6 proteins in the receptor-unbound state is quite complex and is determined not only by the anchoring to the membrane but also by the presence and position of N- and O-glycans and the ability of individual protein regions to interact with the membrane lipids. The relative position and dynamics of the Ly6 proteins weakly depend on the length of the *C*-terminal linker connecting the LU domain with GPI-anchor. GPI-anchoring does not guarantee the optimal pre-orientation of the LU domain required for the receptor interaction. The obtained results are valuable for the ongoing research of the regulatory proteins from the Ly6/uPAR family.

## Figures and Tables

**Figure 1 ijms-24-00011-f001:**
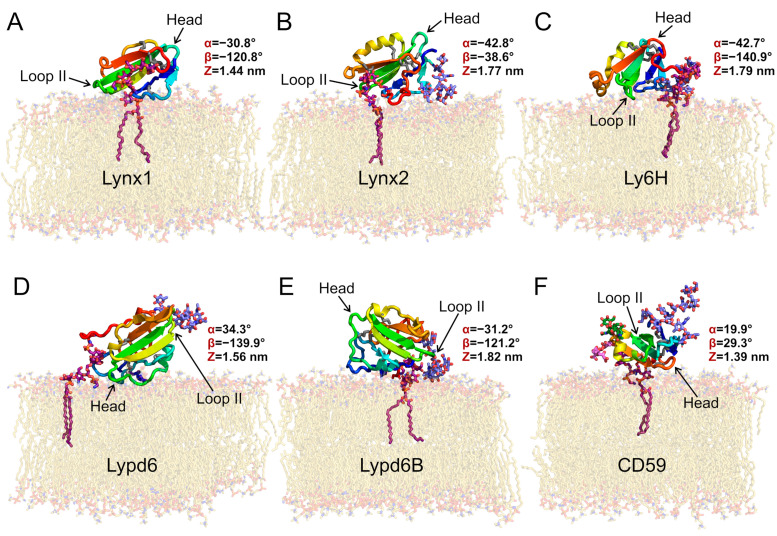
**Preferential orientations of GPI-anchored Ly6 proteins in the membrane:** Lynx1 (**A**), Lynx2 (**B**), Ly6H (**C**), Lypd6 (**D**), Lypd6B (**E**), and CD59 (**F**). Molecules are colored from *blue* (*N*-terminus) to *red* (*C*-terminus); disulfide bonds are shown as *sticks* (sulfur and Cα atoms are *gray*); carbon atoms of GPI-anchors are *purple*; N-glycans are *lavender blue*; O-glucans of CD59 are *pink* and *green*; and lipids are *beige*. All presented proteins except for Lynx1 have N-glycans; CD59 has also two O-glycans. Ly6 proteins adopt different orientations relative to the membrane, forming contacts with membrane via polypeptide fragments and glycans (Table 2). These snapshots represent the most populated clusters from MD (see Figure 4). Variables triad (α, β, and Z) values (see Figure 2 and Appendix A) are given for the most preferential configurations according to MD simulations results.

**Figure 2 ijms-24-00011-f002:**
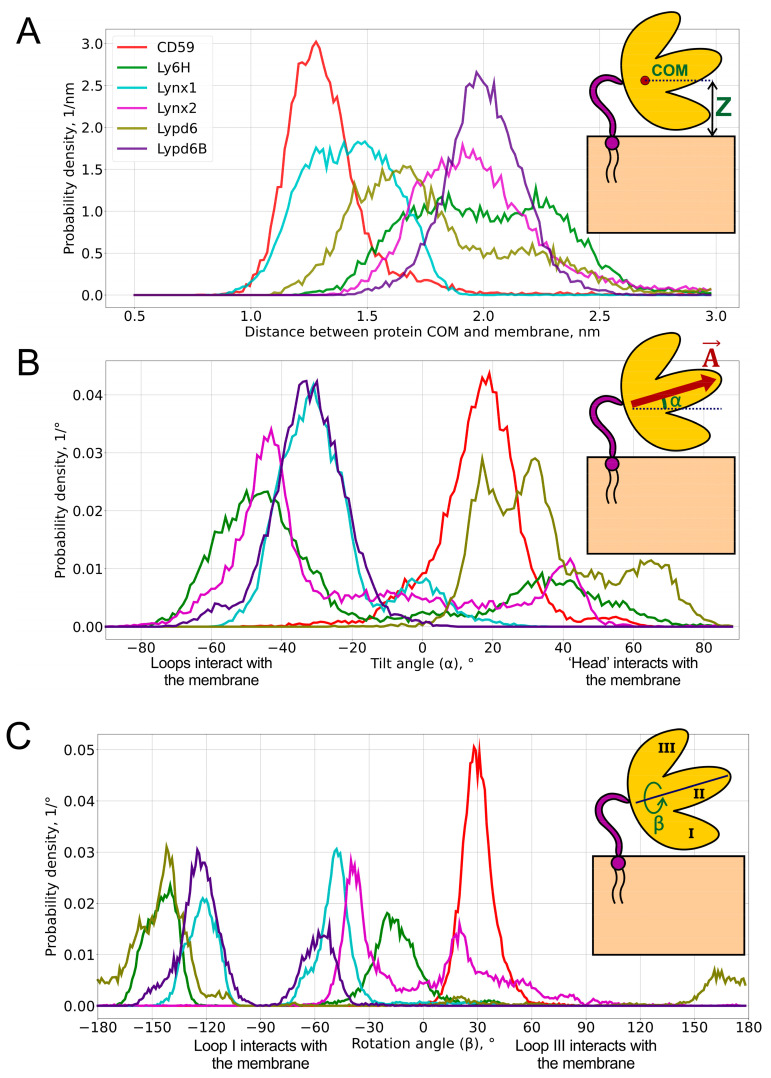
**Probability distributions of the center of mass (COM) position (A), tilt angle α (B), and rotation angle β (C)** of GPI-anchored Ly6 proteins in the membrane. Pictograms describing these parameters are shown as *insets*. Median values and standard deviations of the observed distributions are provided in Table 1. 2D histograms of probability density in α/β-coordinates are represented in Figure 4.

**Figure 3 ijms-24-00011-f003:**
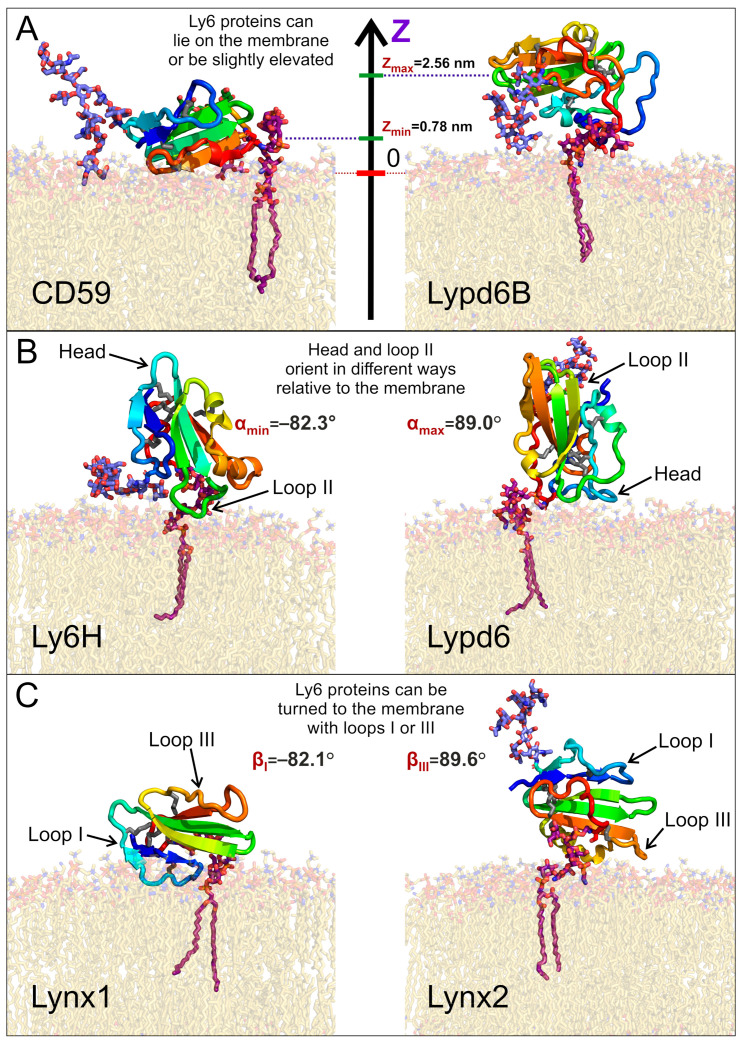
**Diversity of the GPI-anchored Ly6 protein positions (A) and orientations (B,C) relative to the membrane.** Extremal modes with highest and lowest values of the center of mass (COM) position (Z), tilt angle α, and rotation angle β are shown on panels **A**, **B**, and **C**, respectively. Minimal and maximal obtained values are provided. Positions of presented modes in α/β-coordinates are marked in Figure 4.

**Figure 4 ijms-24-00011-f004:**
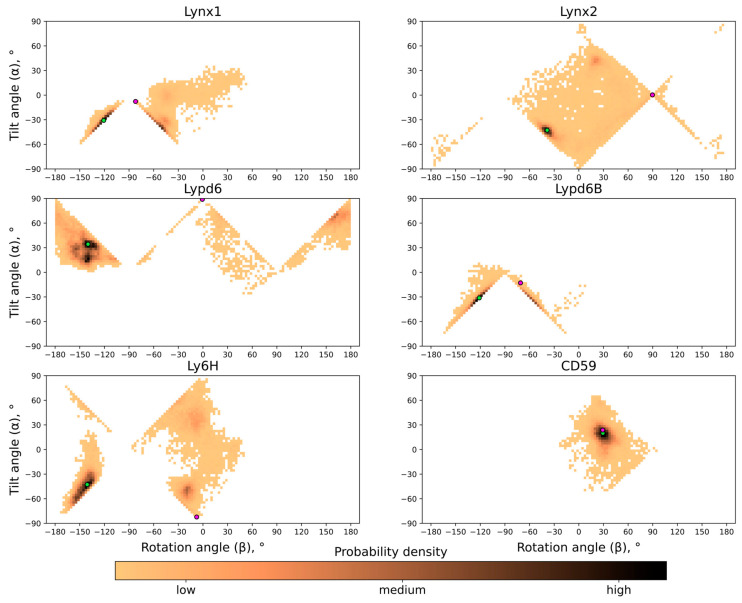
**Probability distribution histograms in tilt (α) and rotation (β) angles’ coordinates.** Green circles correspond to preferential orientations of Ly6 proteins shown in Figure 1; violet circles correspond to extremal orientations shown in Figure 3.

**Table 1 ijms-24-00011-t001:** Positional and orientational preferences of GPI-anchored Ly6 proteins in the membrane.

Protein	Mature Form, Residues *^a^*	Glycosylation Sites	*N*-Terminal Region Length, Residues *^b^*	*C*-Terminal Linker Length, Residues *^c^*	COM Position, nm *^d^*	Tilt Angle α, ° *^d^*	Rotation Angle β, ° *^d^*
Lynx1	21–91	—	0	0	1.43 ± 0.21	–31.5 ± 8.7 (0.7 ± 6.5)	−121.6 ± 8.6 −47.9 ± 7.2
Lynx2	21–117	N: Asn45	2	11	1.93 ± 0.23	−43.8 ± 5.6 (−18.8 ± 37.8) *^e^* (40.9 ± 4.6)	−37.9 ± 5.8 (20.4 ± 4.4) (22.0 ± 38.1) *^e^*
Lypd6	23–147	N: Asn134	24	20	1.62 ± 0.20 (2.23 ± 0.20)	17.3 ± 4.4 31.7 ± 5.9 (60.1 ± 12.3)	−139.9 ± 4.2 −146.2 ± 15.4 (166.6 ± 10.9)
Lypd6B	40–164	N: Asn147	20	24	2.00 ± 0.16	−31.5 ± 9.4	−122.6 ± 9.4 −57.0 ± 8.9
Ly6H	26–115	N: Asn36	0	5	1.78 ± 0.20 2.27 ± 0.18	−46.6 ± 12.0 (37.6 ± 16.1)	−15.7 ± 11.1 −144.0 ± 9.2
CD59	26–102	O: Thr76, Thr77 N: Asn43	0	8	1.29 ± 0.13	17.8 ± 9.0	29.8 ± 7.9

*^a^* The fragment of the protein forming the mature form, from *N*-terminus to the site of the GPI-anchor attachment. Numbering according to the Uniprot database. *^b^* The length of the protein region that precedes the conserved LU domain (which starts with two amino acids before first cysteine). *^c^* The length of the protein linker connecting the conserved LU domain (after the terminal cysteine) with the GPI-anchor. *^d^* The center of mass (COM) position and α/β angles are depicted as *insets* in Figure 2 and explained in the Appendix A. Data are calculated over MD trajectories’ production parts (after energy minimization, NVT-, and NPT-relaxation) and presented as median values ± standard deviations in approximation of uni-, bi-, or trimodal normal distributions. Low-population modes are indicated in parentheses. *^e^* Wide base modes.

**Table 2 ijms-24-00011-t002:** Interactions of GPI-anchored Ly6 proteins with membrane lipids ^a^.

Protein	Ion–Ion and Ion–Dipole Interactions	Hydrogen Bonds	π–Cation Interactions	Hydrophobic Contacts
Lynx1	*Loop I*: D31, R38, *Loop II*: **R57**, **K59** *GPI*: **DSPI-1**, GlcN-2, **PEtN-Man-3**	*Loop I*: Y28, N29, G30, **N32**, C33, F34, **N35** *Loop II*: **Y53**, T54, T56 *Loop III*: Y76	—	*Loop I*: C26, A27, P36 *Loop II*: P55
Lynx2	*Loop I*: E30 *Loop III*: K92 *C-terminus*: R110, K112, K113, **R114** *GPI*: DSPI-1, **GlcN-2**, **PEtN-Man-3**	*Loop I*: Q32 *Loop III*: Y84 *C-terminus*: G115, S116 *N-glycan*: GlcNAc-8	*Loop I*: F31	*Loop I*: N35 *Loop II*: A63
Lypd6	*N-terminus*: R26, K31 *Loop I*: R63, **D67** *Head I*: **R72**, E73 *Head II*: R75 *GPI*: DSPI-1, **GlcN-2**, **PEtN-Man-3**	*N-terminus*: Y43, G46 *Loop I*: W64, **Y69** *GPI*: Man-6	*Loop I*: W64, Y69	*N-terminus*: P44, G45, K48 *Loop I*: P66, I68 *Head I*: P71 *GPI*: Man-5
Lypd6B	*Loop I*: **R76** *Loop II*: R101 *GPI*: DSPI-1, **GlcN-2**, **PetN-Man-3**	*N-terminus*: **Y45**, N46 *Loop I*: Y72, N73, W77 *N-glycan*: Man-4, GlcNAc-6, GlcNAc-8	*Loop I*: W77	*N-terminus*: N43, V47, P49, P50 *Loop I*: D70 *C-terminus*: S164 *GPI*: Man-5
Ly6H	*Loop II*: **R64**, **K65** *Head III*: K107 *GPI*: DSPI-1, **GlcN-2**, **PEtN-Man-3**	*Loop I*: T35 *Loop II*: S63 *Loop III*: Y88 *N-glycan*: GlcNAc-5, Man-7, GlcNAc-8 *GPI*: Man-4, Man-5	*Loop I*: H39	*Loop III*: F92 *N-glycan*: GlcNAc-1
CD59	*Head III*: **K90**, **K91** C-terminus: E101 *GPI*: GlcN-2, PEtN-Man-3	*Loop III*: Y86, Y87 *C-terminus*: Q99, N102 *O-glycan (T76)*: Neu5Ac-3 *GPI*: **SAPI-1**	*Head II*: F67	*Head II*: E68, C70 *Loop III*: N71, F72, N73 *Head III*: C88, C89, C94 *C-terminus*: F96, E98, L100 *O-glycan (T76)*: Gal-2 *GPI*: Man-5

^a^ Table includes amino acids, carbohydrates, and phosphatidylinositol residues interacting with membrane lipids for at least 10% of the MD time through ion–ion/ion–dipole interactions and hydrogen bonds; for at least 5% of the MD time through π–cation interactions; or forming at least two hydrophobic contacts with a total lifetime of more than 200% of the MD trajectory (100% corresponds to one contact during the whole trajectory). Residues that simultaneously form the ionic contacts and hydrogen bonds are listed in the first column only. The column with hydrophobic contacts lists only the residues that do not participate in the ionic and π–cation interactions or hydrogen bonds. Strong ionic interactions and hydrogen bonds with a total lifetime of ≥ 50% of the MD time are shown in *bold*; the most stable of them (with lifetime ≥ 75%) are also *underlined*. The Loop/Head I/II/III regions of the proteins are defined in Appendix A. Designations: SAPI—stearoylarachidonoylphosphatidylinositol; DSPI—distearoylphosphatidylinositol; and PEtN—phosphatidylethanolamine.

## Data Availability

MD trajectories, Appendix A are available on Zenodo: https://doi.org/10.5281/zenodo.7074188 (accessed on 15 December 2022).

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
