# Peer review of "Orientational Preferences of GPI-Anchored Ly6/uPAR Proteins"

_ijms, 2022, doi:10.3390/ijms24010011_

Round 1

Reviewer 1 Report

In this manuscript, the author studied the orientational preference of GIP-anchored Ly6/uPAR proteins. This is an interesting and useful work. I have following comments.

1.     GPI should be defined in the title and the abstract.

2.     One usually uses three numbers to describe the position of the COM and three angles to describe the orientation of a rigid body. In this work, the author only used z, alpha, beta. While I understand this is because there are translational and rotational symmetries with respect to the membrane, the authors should explain this to avoid any confusions.

3.     Is the structure of the proteins allowed to relax (change) during the MD simulations?

4.     What is the energy difference for different orientations? For example, in Figure 2a, can the authors plot energy vs. distance for several fixed alpha and beta? This information should be useful.

Reviewer 2 Report

Attached file.

Round 2

Reviewer 1 Report

The authors have addressed my conerns satisfactorily. I suggest publication.

Author Response

We thank the Reviewer for his/her time and consideration, and are glad that this paper may proceed now to the publication.

Reviewer 2 Report

Attached file.

Author Response

We thank reviewer for his/her attention and positive attitude to our work, and made the recommended minor corrections:

  1. Transferred the Fig. S6 ("FEP plot") to the main text as new Fig. 4, while left Fig. S2 ("PCA plot") "as is", since visually it's very similar to the Fig. 1, and it would be not very gentle to the reader to put two such alike pictures side-by-side.
  2. Also, we discussed a bit more the two mentioned results (PCA/FEP).
  3. As for pictures quality, they're submitted to the journal with maximal quality (4K), but for the review purpose the compressed file is produced. We hope that final "production" pdf will be available in the optimal quality/resolution.